# The BRCA Gene in Epithelial Ovarian Cancer

**DOI:** 10.3390/cancers14051235

**Published:** 2022-02-27

**Authors:** Luisa Sánchez-Lorenzo, Diego Salas-Benito, Julia Villamayor, Ana Patiño-García, Antonio González-Martín

**Affiliations:** 1Department of Medical Oncology, Clínica Universidad de Navarra, 28027 Madrid, Spain; lsanchezl@unav.es (L.S.-L.); dsalas@unav.es (D.S.-B.); jvillamayor@unav.es (J.V.); apatigar@unav.es (A.P.-G.); 2Solid Tumor Program (CIMA), Clínica Universidad de Navarra, 31008 Pamplona, Spain

**Keywords:** BRCA, PARPi, epithelial ovarian cancer, hereditary breast and ovarian cancer

## Abstract

**Simple Summary:**

Epithelial ovarian cancer (EOC) is still the most lethal gynecological cancer. In the recent years, the germline alterations in breast cancer 1 (gBRCA1) and breast cancer 2 (gBRCA2) genes are of key importance not only for genetic counseling purposes, but for its therapeutic implications, as well as somatic mutations, for the latter. The integration of poly-(ADP ribose) polymerase inhibitors (PARPis) as part of the therapeutic options has changed EOC natural history.

**Abstract:**

Epithelial ovarian cancer (EOC) is still the most lethal gynecological cancer. Germline alterations in breast cancer 1 (gBRCA1) and breast cancer 2 (gBRCA2) genes have been identified in up to 18% of women diagnosed with EOC, and somatic mutations are found in an additional 7%. Testing of BRCA at the primary diagnosis of patients with EOC is recommended due to the implications in the genomic counseling of the patients and their families, as well as for the therapeutic implications. Indeed, the introduction of poly-(ADP ribose) polymerase inhibitors (PARPis) has changed the natural history of patients harboring a mutation in BRCA, and has resulted in a new era in the treatment of patients with ovarian cancer harboring a BRCA mutation.

## 1. BRCA Gene as Therapeutic Target

### 1.1. DNA Damage Repair (DDR) and BRCA Function

The maintenance of genomic stability is crucial for DNA integrity and cell survival; thus, the proteins involved in the different repair pathways are usually involved in various cellular processes in response to DNA damage, and their targeting may be used as a therapeutic approach against tumors and/or patients carrying such alterations [1].

The human cells (in general all eukaryotic cells) developed different pathways to fix the different types of DNA damage, either affecting one (SSBs, single strand breaks) or both (DSBs, double-strand breaks) DNA strands. When only one DNA strand is broken, and thus the other is available as a template, it can be repaired by base excision repair (BER), nucleotide excision repair (NER) or mismatch repair (MMR) [2]. BER corrects the forms of single base damage that are not perceived as a significant distortion to the DNA helix. NER, instead, repairs multiple and bulky base damage, while MMR is an evolutionarily conserved, post replicative repair pathway, which contributes to replication fidelity [3]. If both strands are involved, the pathways available for repair are non-homologous end joining (NHEJ) and homologous recombination (HR), which includes gene conversion (GC) and single strand annealing (SSA) [2]. NHEJ is critical for the repair of pathologic DSBs, chromosomal translocations and also for the repair of physiologic DSBs created during V(D)J recombination and class switch recombination. HR utilizes DNA strand invasion and template-directed DNA repair synthesis to effect a high-fidelity repair [3].

BRCA1 and BRCA2 proteins are mediators of genome integrity, mainly through HR DSB repair. Thus, both patients with *BRCA1* or *BRCA2* germline mutations or somatic inactivation (one or usually both alleles) can benefit from targeted treatments, such as the poly (ADP-ribose) polymerase (PARP) inhibitors (PARPi) that lead to increased DSB in cells with HR impairment. PARP is a family of enzymes that have a key role in BER repair in response to SSBs; when PARP is inhibited, the DNA breaks persist and result in potentially permanent DNA damage. Treatment of BRCA mutant tumors with PARPi leads to DNA damage accumulation and, thus, to cell cycle arrest and cell death. This effect of PARP inhibition in cells with a mutant HR pathway is known as synthetic lethality (Figure 1) [4]. PARP activity is also useful for tumor cells to avoid death due to chemotherapy, which occurs most frequently by the BER pathway. Thus, when PARP is inhibited (by PARPi) the tumor cells have increased sensitivity to certain cytotoxic agents.

The term BRCAness was initially used to describe a phenotype consistent with a defect in the HR repair, such as *BRCA1/2* alteration. Most recently, the definition is being broadened to include replication fork protection mechanisms as well as other pathways that lead to synthetic lethality with PARPi. Thus, BRCAness may lie in aberration of the genes *BRCA1, BRCA2, ATM, BARD1, BRIP1, CHEK1, CHEK2, FAM175A, MRE11A, NBN, PALB2, RAD51C* and *RAD51D*, and, in ovarian cancer, BRCA1 or BRCA2 mutation predicts the high sensitivity to platinum-based chemotherapy and PARPi, leading to an increased overall survival [5].

### 1.2. Challenges of BRCA Testing: Germline vs. Somatic vs. Both

Since both germline and somatic *BRCA1/2* gene mutation are predictive factors to respond to PARPi, an important question of BRCA testing as a predictive factor is whether the analysis should be initiated by germline mutations in the blood or by testing somatic mutations in the tumor tissue. This decision will also determine the technical recommendations and limitations in the pre-analytical (sample processing and nucleic acid extraction and quantification), analytical (NGS, next generation sequencing analysis) and post-analytical (QC, quality control, analysis and filtering of variants) (Table 1) [6].

Finally, the increasing use of BRCA1/2 testing in somatic tissues for tailoring cancer treatment, in addition to classical germline testing for inherited predisposition diagnosis is switching the classical prevention scenario to mainstream oncology practice. In this “new” practice, with an increasing number of genes being analyzed as the definition of BRCAness broadens, a significant number of genetic variants without a clear clinical significance (VUS, variants of unknown significance) will be identified and their correct interpretation is crucial for treatment recommendation and genetic counseling [7].

## 2. BRCA Gene and Hereditary Ovarian Cancer Syndrome

Current guidelines from different scientific societies, such as NCCN, ASCO, ESMO-ESGO and SGO, strongly recommend the genetic testing of BRCA1/2 for every newly diagnosed patient with a non-mucinous epithelial ovarian cancer (which includes fallopian tube and primary peritoneal cancers) regardless of family history. 

Hereditary breast and ovarian cancer (HBOC) caused by germline BRCA1/2 pathogenic mutations (gBRCAms) is predicted to be responsible for about 5% of breast cancers as well as the 15–18% of all ovarian cancers, and an additional 5–7% show somatic BRCA1/2 pathogenic mutations (sBRCAms) [8,9].

Germline pathogenic variants in BRCA1/2 are inherited in an autosomal dominant pattern, such as the transmission of classic tumor suppressor gene mutations, and most individuals have inherited it from a parent. Penetrance by the age of 70 years for breast cancer in BRCA1 mutation carriers has been estimated to be 64.6% (95% confidence interval (CI), 59.5–69.4%) and 61.0% (95% CI, 48.1–72.5%) in BRCA2 mutation carriers, meanwhile for ovarian cancer it is 48.3% (95% CI, 38.8–57.9%) for BRCA1 and 20.0% (95% CI, 13.3–29.0%) for BRCA2 mutation carriers [10]. BRCA mutation carriers are also at risk, although to a lesser extent, of other malignancies, such as melanoma, endometrial, pancreatic, prostate and colorectal cancer [11]. 

Although several founder mutations of the *BRCA1* and *BRCA2* genes have been reported, the breast and ovarian cancer risks varied by type and location of BRCA1/2 mutations. An example of this is the identification of an ovarian cancer cluster region (OCCR) in or near exon 11 in *BRCA1* and *BRCA2*. The Consortium of Investigators of Modifiers of *BRCA1/2* (CIMBA) revealed that the incidence of OC is high in patients with a germline BRCA mutation in the OCCR in about 30,000 BRCA mutant carriers in 33 countries around the world. Pathogenic variants within the OCCR have been associated with a higher ratio of ovarian to breast cancer [12]. 

Rebbeck et al., using the largest dataset analyzed, identified in BRCA1 an OCCR1 from c.1380 to c.4062 (approximately exon 11), and 2 OCCRs were identified in BRCA2: the OCCR1 spanned from c.3249 to c.5681 (exon 11), adjacent to c.5946delT (exon 11) and the OCCR2 spanned from c.6645 to c.7471 (between exon 11 and 15) [12].

When a somatic BRCA1/2 pathogenic or likely pathogenic mutation is detected, based on current norms used by available genetic testing platforms, the patient must be referred to the Genetic Counseling Unit for germline genetic testing. Relying on the variant allele frequency is not recommended for making decisions about whether a given variant may be in the germline or not, as it can be affected by tissue heterogeneity, tumor heterogeneity or copy number abnormalities [13]. 

The identification of pathogenic variant carriers and at-risk individuals is of the utmost importance as it may reduce morbidity and mortality from cancer. Genetic testing for those at risk of identifying a pathogenic variant before a diagnosis of cancer allows for the consideration of advanced surveillance diagnostics, therapeutics, or surgical interventions within a process of genetic counseling [14]. 

There is an increased risk of cancers of the ovary, fallopian tube, and peritoneum in BRCA1/2 mutation carriers. Those patients are classically diagnosed with high-grade serous adenocarcinomas with a frequency ranging from 67 to 100%, albeit endometrioid and clear cell ovarian cancers have also been reported with a frequency similar to the general population [15,16]. Other tumor types have been described, accounting for <10% of all tumors. High grade serous adenocarcinomas are more aggressive and with poorer prognosis, as highlighted by morphological and ultrastructural studies [17]. Current data show that ovarian cancer with a low malignant potential (borderline epithelial ovarian tumor) is not associated with BRCA1/2 mutations [14]. The mucinous ovarian cancers and non-epithelial carcinomas as germ cell and sex cord stromal tumors are not significantly related to these mutations [18].

The prevalence of ovarian cancer in BRCA1 mutation carriers is 1.5% in <40 years of age and increases up to 10–21% by 50 years of age, while in BRCA2 mutation carriers, the risk is less than 3–5% by 50 years of age [19,20]. Based on this, the international guidelines recommend risk-reducing bilateral salpingo-oophorectomy (BSO) once childbearing is complete, ideally between the ages of 35 and 40 years for BRCA1 and between the ages of 40 and 45 years for female carriers of BRCA2 mutations. The prophylactic BSO reduces ovarian cancer incidence up to 79–83% and breast cancer incidence up to 50% [21]. This translates to a reduction in ovarian cancer-specific mortality between 80–96% and in breast cancer-specific mortality of 42%, which is more pronounced for BRCA1 (HR 0.45, *p* < 0.0001) vs. BRCA2 mutation carriers for whom the reduction loses significance (HR 0.88, *p* = 0.75) [22]. A recent meta-analysis, including 13,871 BRCA1/2 mutation carriers, evaluating the prevalence of endometrial cancer reported a slightly increased risk, mainly for BRCA1, but the absolute risk remained low [23]. When discussing hysterectomy with the patient, we should bear in mind the patient’s age, type of mutation, need for hormone replacement treatment, history of breast cancer, tamoxifen use, and individual surgical risks.

About 3–10% of BRCA1/2 mutation carriers who undergo BSO are diagnosed with occult fallopian tube and ovarian cancers. Considering this risk, consensus guidelines recommend a complete pathology review and serial sectioning of the ovaries and fallopian tubes to exclude occult cancers or serous intraepithelial tubal carcinomas (STICs) [24]. According to the actual guidelines, adjuvant chemotherapy is not recommended in the management of incidentally detected isolated STIC lesions [25]. 

For women who refuse BSO or postpone it until after the natural menopause, a combination of transvaginal ultrasounds and serum Ca125 every 6 months may be considered from the age of 30, although the benefit of these tools has not been established [26]. 

## 3. BRCA Gene in the Clinic

### 3.1. Prognostic Implication

Chemosensitivity to platinum-based regimens in ovarian cancer patients has been associated with a better prognosis. The cellular mechanism that makes platinum so efficient at stopping and collapsing the cell cycle has been studied for years. The main mechanism to restore the DNA integrity and cell survival is the homologous recombination, which includes a complex protein machinery that involves BRCA1 and BRCA2 proteins. The failure of homologous recombination does not allow for the double-strand break correction induced by platinum compounds. In this situation, those patients harboring mutations in these genes could have a higher sensitivity to platinum than those with a wild-type status of these proteins [27]. In fact, patients considered to have platinum-resistant disease, according to the old definitions based on progression within six months after completing the last platinum-based chemotherapy, and BRCA1/2 mutation, may respond to a platinum rechallenge [28]. By contrast, some platinum resistance mechanism involved HR restoration, leading to a more efficient repair system that diminished platinum-induced lethality.

It is worth noting that platinum sensitivity advocates better outcomes with other chemotherapeutic agents, which are also related to the HR repair system, such as topotecan or doxorubicin. Moreover, the combination of platinum with gemcitabine demonstrated good results between patients with platinum-resistant disease due to a synergistic effect blocking the carboplatin adduct intra-strand repair machinery [29].

Many studies demonstrated that germline and somatic BRCA1 and/or BRCA2 mutations are related to a better prognosis in ovarian cancer patients. This benefit was demonstrated both in ovarian and breast cancers. For ovarian cancer, the benefit of BRCA1/2 mutation in overall survival (OS) and in progression free survival (PFS) was not related to stage, histology subtype or grade. In the Australian Ovarian Cancer Study Group analysis, the OS and PFS of the patients with BRCA1/2 mutations in the germline were higher than in those patients without these genetic alterations. The only independent factor related to a better outcome in the mutant group was an optimal debulking surgery. In fact, patients with BRCA1/2 mutations and suboptimal debulking surgery had a similar outcome than those with the wild-type genotype but optimal surgery [28]. This observation was confirmed in a recent meta-analysis of the MITO Italian group describing the features of patients with BRCA1/2-mutated EOC, showing that the variable that better correlated with the outcomes (OS and PFS) was a complete cytoreduction surgery, suggesting that surgery continues to be a milestone in the treatment of patients with EOC, irrespective of BRCA mutation status [30].

In a meta-analysis conducted by Zhong Q et al. from 14 studies of patients with newly diagnosed epithelial ovarian cancer (EOC), the hazard ratios (HR) for PFS and OS were 0.65 (CI 95%; 0.52–0.81) and 0.76 (CI 95%; 0.70–0.83), respectively, in BRCA1 mutant patients. For those harboring BRCA2, the PFS and OS hazard ratios were 0.61 (CI 95%; 0.47–0.80) and 0.58 (CI 95%; 0.50–0.66), respectively [31]. Chemosensitivity in BRCA mutated EOC was also observed when compared with sporadic EOC in terms of complete response or no evidence of disease (87% in BRCA1, 92% in BRCA2 and 71% in sporadic cases). If we compare the specific benefit of BRCA1 and 2 mutations, it seems that BRCA2 mutated EOC patients have a better outcome than the patients harboring the BRCA1 mutation [32]. The clinical effects of BRCA1 and BRCA2 mutations are commonly analyzed together, but several studies found that, compared with BRCA1 mutation carriers, BRCA2 mutation carriers were associated with an improved platinum-based chemotherapy response and longer PFS [33,34,35].

### 3.2. Predictive Factor of PARPi Sensitivity

The polyadenosine diphosphate ribose polymerase (PARP) is an enzyme participating in the single-strand DNA break repair system called base excision repair (BER). The sensitivity of tumors with homologous recombination deficiency to PARP inhibitors was demonstrated in many studies (see below). Numerous inhibitors have been developed during the last years, inducing a blockade of this DNA repair system that, in tumors harboring BRCA1/2 mutations, lead to a double-hit in the DNA repair machinery (single- and double-strand damage), and finally the apoptosis of the cell due to an arrest in cell cycle and collapse. This effect has been called synthetic lethality. This concept was introduced a century ago by geneticists explaining that the combinations of defects in various genes has a higher deleterious effect than it has separately.

However, it continues to be not well known that the exact mechanism by PARPi induces its effect in tumors cells with HRD. It has been postulated that there are many pathways affected by PARPi. Base excision repair (BER) depends on PARP1 function, and the inhibition of this protein by PARPi in a cell with DNA DSB machinery repair knock-out will induce a cell collapse and apoptosis. Additionally, blocking PARP1 will promote classic nonhomologous end-joining (C-NHEJ) that will increase genome instability in HRD cells, and finally synthetic lethality; the connection between C-NHEJ PARPi-mediated activation and HRD/PARPi synthetic lethality must be elucidated. PARPi can also “trap” PARP1 on the DNA, preventing the autoPARilization and excluding PARP1 from the site of DNA damage; in fact, cell knock-out for PARP1 is resistant to PARPi. Other mechanisms that confer a higher sensitivity to PARPi are the alternative end-joining repair defect and the inhibition of all the family of PARP proteins [36].

Interestingly, a meta-analysis of Li. S. et al. from 11 randomized controlled trials of PARPi in BRCA1/2-mutated populations, including ovarian, breast, pancreatic and prostate cancer, was performed to assess the efficacy difference of PARPi between BRCA1 and BRCA2 mutation carriers. The pooled PFS HR was 0.42 (95% CI: 0.35–0.50) in BRCA1 and 0.35 (95% CI: 0.24–0.51) in BRCA2 mutation carriers compared with patients in the control group. The difference in efficacy of PARPi was not significant between the two subgroups (P_heterogeneity_ = 0.40 for interaction). Even in the subgroup analyses performed according to the cancer types, study methodology and lines of therapy, no statistically significant differences in the efficacy of PARPi were found between the BRCA1- or 2-mutated patients. Both BRCA1 mutation carriers and 2 mutation carriers could significantly benefit from PARPi regardless of cancer types and therapeutic lines [37]. 

A possible explanation for this lack of difference might be related to other therapeutic actions of PARPi and other functions of *BRCA1* beyond DNA damage. *BRCA2* guides RAD51 to damage sites in the process of DNA repair. Meanwhile, *BRCA1* in addition to DNA repair also plays a critical role in checkpoint control, mitotic spindle assembly, sister chromatid decatenation and centrosome duplications [38]. 

## 4. Clinical Data with PARPi in BRCA-Mutated OC Patients

The incorporation of PARP inhibitors has dramatically changed the landscape of the treatment for patients with advanced ovarian cancer. Although they have shown to be effective as maintenance therapy after platinum-based chemotherapy response regardless of biomarkers, both in the recurrent setting (NOVA/ENGOT-Ov16, NORA and ARIEL-3 studies) and the front line (PRIMA/ENGOT-Ov26/GOG 3012); the highest benefit is achieved in patients harboring a BRCA mutation making BRCA the more accurate biomarker of response to PARP inhibitors.

### 4.1. Maintenance Therapy

PARP inhibitors were first evaluated as maintenance therapy in patients with recurrent ovarian cancer that achieved a partial or complete response to a platinum-based chemotherapy rechallenge. Table 2 summarizes the most relevant studies as well as the specific data for patients with BRCA mutation in the recurrent setting. It is worth mention that the randomized phase II clinical trial (Study-19) was the proof of concept, indicating that the addition of Olaparib as maintenance after response to platinum in recurrent disease significantly prolongs the median progression-free survival from 4.8 to 8.4 months in the whole population (HR 0.35; 95% CI, 0.25 to 0.49; *p* < 0.001) [38]. Interestingly, an exploratory analysis in the patients with germline or somatic BRCA mutations that represented roughly half of the population of the study, showed an unprecedent increment in PFS of 6.9 months with a HR 0.18 (CI 95%, 0.10–0.31; *p* < 0.0001) and a number of long-term responders without progression after more than 5 years of treatment [39]. These clinically impactful data were confirmed in the SOLO-2/ENGOT-Ov 21 trial, in which 327 patients with platinum-sensitive recurrent disease responding to platinum rechallenge were randomized 2:1 to Olaparib 300 mg bid tablets of placebo. SOLO-2 not only confirmed that Olaparib significantly prolonged the PFS for 13.6 months with a HR 0.30 (95% CI, 0.22–0.41, *p* < 0.0001) [40], but it also showed an increment in overall survival, despite a cross-over rate of 38.4%, which was not statistically significant (HR 0.74; 95% CI, 0.54–1.00; *p* = 0.054), but was clinically relevant with an increment of 12.9 in median OS [41].

The NOVA/ENGOT-Ov16 study was the first randomized phase III trial indicating a benefit of PARPi as maintenance regardless of BRCA status. However, the benefit in the cohort of gBRCAmut was more pronounced than in the non-gBRCAmut cohort [42]. These data were reproduced in a phase III randomized clinical trial named NORA that included an only-Chinese population [43]. Finally, ARIEL-3 included patients with recurrent ovarian cancer responding to platinum that were randomized to Rucaparib or placebo, demonstrating in the hierarchical analysis the highest benefit for patients with BRCAmut tumors [44].

Four trials analyzed the impact of PARPi as maintenance after front-line platinum-based chemotherapy (Table 3). SOLO-1 with Olaparib was restricted to patients with a BRCA mutation [45]; the PRIMA/ENGOT-Ov26 study with Niraparib was compared to placebo for all comers regardless of biomarker status [46]; the PAOLA-1/ENGOT-Ov25 trial with Olaparib was added to bevacizumab compared to bevacizumab [47], and VELIA with veliparib was added to chemotherapy followed by maintenance with veliparib [48]. In PRIMA and PAOLA-1, patients harboring a BRCA tumor mutation were analyzed as a pre-planned exploratory endpoint, but in VELIA this population was analyzed first following the hierarchical testing analysis. As it was observed in the recurrent setting, the BRCA mutant population achieved the highest benefit from PARP-inhibitor maintenance in the front line as summarized in Table 3. Of note, patients with a HR deficiency, but BRCA wild-type defined as GIS scoring > 42 determined by Myriad Mychoice, also obtained a significant improvement in PFS with PARP inhibition in PRIMA and PAOLA, but not in VELIA. Finally, only Niraparib showed a benefit in patients with HR-proficient tumors. The technical aspects for assessment and the clinical utility of HR testing is beyond the scope of this manuscript and was well-addressed in other excellent reviews [49].

Importantly, the long-term result of the SOLO-1 trial with a follow-up of more than 5 years has shown that the hazard ratio is maintained, and the survival curves stand separated, with approximately 50% of patients remaining without progression, which probably means that many of those patients have been cured due to the incorporation of PARP as maintenance [50].

Based on the above-mentioned evidence, all patients with a germline or somatic BRCA mutation should receive a PARP inhibitor as maintenance therapy after front-line chemotherapy, if they have not progressed on it. Niraparib alone, Olaparib alone, or the combination of Olaparib and bevacizumab are the current options in the clinic for BRCAmut patients according to the approvals by the FDA, EMA and many other health authorities.

### 4.2. PARPi Single Agent as Treatment 

In addition to the approvals of PARP inhibitors as maintenance after platinum-based chemotherapy in the front line and the recurrent setting, we also have the approval of PARP inhibitors as monotherapy for the treatment of patients with BRCA mutant tumors (Table 4). Olaparib was approved by the FDA for patients with recurrent ovarian cancer who have received 3 or more lines of chemotherapy and harbor BRCA mutations based on a phase 2 trial [51]. The value of Olaparib monotherapy in this setting was confirmed in the SOLO-3 trial, a phase 3 clinical trial that compared Olaparib 300 mg twice a day or physician’s choice single-agent nonplatinum chemotherapy (pegylated liposomal doxorubicin, paclitaxel, gemcitabine, or topotecan) in patients with BRCA-mutated tumors and platinum-sensitive relapse who received at least 2 prior lines of platinum-based chemotherapy. The primary end-point was the objective response rate (ORR) that was significantly higher with Olaparib than with chemotherapy (72.2% vs. 51.4%; odds ratio (OR), 2.53 (95% CI, 1.40 to 4.58); *p* = 0.002). This benefit was restricted to patients with 2 to 3 prior lines versus more than 4, and was more clear for patients with a platinum-free interval from 6 to 12 months versus more than 12 months [52].

Rucaparib is approved as monotherapy for patients with BRCA-mutant tumors who received 2 or more lines of chemotherapy regardless of platinum-free intervals based on the results of ARIEL-2 and Study-10 clinical trials [53]. The ARIEL-4 trial enrolled 349 patients with high-grade epithelial ovarian cancer and a germline (84%) or somatic (16%) BRCA1/2 mutation, who received two or more prior chemotherapy regimens. Patients were randomly assigned to receive Rucaparib 600 mg twice daily or standard-of-care chemotherapy consisting of weekly paclitaxel for patients with a platinum-free interval (PFI) of less than 12 months or investigator’s choice of platinum-based chemotherapy for patients with fully platinum-sensitive disease (PFI ≥ 12 months). Median progression-free survival was 7.4 months with Rucaparib and 5.7 months with chemotherapy (hazard ratio (HR) = 0.64, *p* = 0.001). The objective response rate was similar between the treatment arms: 40.3% in the Rucaparib arm and 32.3% in the chemotherapy arm (*p* = 0.13) [54]. 

Lastly, based on QUADRA clinical trial results [55], Niraparib was approved by the FDA for the treatment of adult patients with advanced ovarian, fallopian tube, or primary peritoneal cancer who were treated with three or more prior chemotherapy regimens and whose cancer was associated with a homologous recombination deficiency (HRD)-positive status defined by either a deleterious or suspected deleterious BRCA mutation, or genomic instability and who progressed more than six months after a response to the last platinum-based chemotherapy.

### 4.3. Challenges and Future Approaches

Despite the high efficacy of PARP inhibitors as maintenance after front-line chemotherapy in patients with BRCA-mutated tumors, approximately half of the patients will relapse in the first 5 years due to primary or secondary resistance to PARPis. The main mechanisms of acquired resistance to PARP inhibitors include the restoration of the homologous recombination (HR) double-strand DNA break repair system, alteration in the trapping of the PARPi enzyme, replication fork stability and PARPi efflux by the p-glycoprotein pump in the multidrug resistance (MDR) system. BRCA reversion mutations that are present in approximately 20% of cases of EOC are the most frequent cause of HR restoration. Other forms of restoration of HR take place in other HR proteins, such as RAD1C and RAD51D. Other mechanisms are the alteration in the cellular cycle; this includes the overexpression of cycle cell regulators, such as CDK-12 and WEE1, alterations in the pattern of expression of miRNA and dysregulation of signal pathways, such as MET, PI3K/AKT and ATM/ATR. A thorough analysis of the mechanism of resistance in not under the scope of this review and the reader is referred to recent comprehensive papers on this topic [36,56].

From a clinical perspective, the issue of PARPi resistance was recently illustrated with the results of the OREO/ENGOT-Ov 38 trial [57]. In this trial, patients with non-mucinous platinum-sensitive relapse, one prior line of PARPi maintenance and in response to their most recent platinum-based chemotherapy were randomized (2:1) to Olaparib (300 mg bid or 250 if 300 not previously tolerated) or placebo until progression. Patients enrolled in BRCA1/2 mutated (BRCAm) were required to have a prior exposition to PARPi ≥ 18 months in the first line or ≥12 m in the second or more lines. In this BRCAm cohort, the median PFS was 4.3 with Olaparib versus 2.8 months with placebo (hazard ratio (HR) 0.57; 95% CI 0.37–0.87; *p* = 0.022). Although the study was statistically positive, it reported that almost 50% of patients progressed in the first 3 months without differences, in comparison with the placebo, showing that those patients were PARPi resistant.

With the aim of overcoming the resistance to PARPi, several combinations were explored in preclinical models, and some of them were evaluated or are under evaluation in clinical trials. It is worth mentioning the combination of anti-angiogenic agents, immune check-point inhibitors, PI3K inhibitors, MEK inhibitors, ATR/Chk1/Wee1 inhibitors and BET inhibitors that are at different stages of development [36].

## 5. Conclusions

The *BRCA* gene plays a critical role in the pathogenesis of a significant number of patients with epithelial ovarian cancer and is a key element in the homologous recombination system for DNA double-strand break repair. Due to implications in the genetic counseling for the patients and their families, as well as its crucial role in the therapeutic decision-making process, testing the BRCA status (both germline and somatic) is recommended in every patient with newly diagnosed non-mucinous ovarian cancer. For patients with BRCAmut tumors, the incorporation of PARP inhibitors as maintenance after response to front-line platinum-based therapy has changed the natural history of these patients and is currently considered as the standard of care. The understanding of the primary and secondary mechanisms of resistance to PARP inhibitors, especially in BRCAmut tumors, is currently one of the most relevant research challenges in ovarian cancer. Finally, many ongoing trials will eventually determine if a combination of PARP inhibitors with other targeted therapies or immunotherapy will improve the outcome of patients with BRCA-mutated tumors.

## Figures and Tables

**Figure 1 cancers-14-01235-f001:**
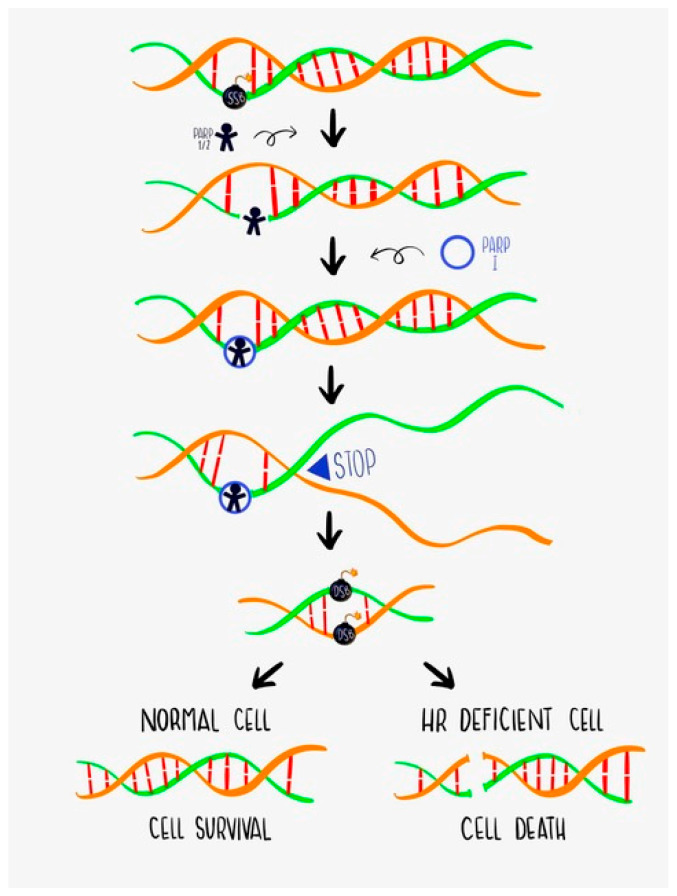
PARPi and synthetic lethality. Single-strand DNA breaks (SSB) lead to PARP1/2 recruitment and activation; PARPi is retained in DNA and results in replication fork stalling. The resulting double-strand DNA breaks (DSB) require homologous recombination (HR) pathway integrity for repair; thus, HR-deficient (BRCA mutant) cells are unable to DNA repair, resulting in cell death.

**Table 1 cancers-14-01235-t001:** Technical considerations for the somatic and germline analysis of HR mutations.

	Germline HR Testing	Somatic HR Testing
Sample	Blood (EDTA anticoagulated) or derivative fractions	Fresh or FFPE tissue, cytological samples
	Processing is not very good	Important fixation, paraffination, decalcification conditions
	Rapid sample processing to avoid RNA degradation	Sample selection for at least 30% tumor cells (>50% ideally)
Analytical considerations	Only germline variants are detected	Both germline and somatic variants are detected
	Expected VAF, around 50%	Expected VAF from 5%
	Read depth 50× to 200×	Recommended read depth 500× to 2000×
	No variants due to technical issues	Potential false positives due to fixation
	More straightforward and validated NGS and pipeline analyses	Complex and difficult to implement NGS and pipelines
	10% of patients with indication will be missed (somatic only)	All patients with indication will be detected
Other considerations	False negative results at homopolymeric traits with mutations	
	Significant number if VUS identified	
	Need for accurate detection of large genomic rearrangements and CNVs	

CNVs, copy number variants; FFPE, formalin-fixed paraffin-embedded tissue; HR, homologous recombination pathway; NGS, next generation sequencing; VAF, variant allele frequency; and VUS, variants of unknown clinical significance.

**Table 2 cancers-14-01235-t002:** Clinical trials of PARP inhibitors as maintenance in the recurrent setting.

Study	Phase	Population	Study Arm	Control Arm	Results
Study-19 NCT00753545	II	Recurrent HG (G2 or 3) OC/FP/PPC ≥2 platinum-based chemotherapy With an objective response to the platinum regimen	Olaparib 400 mg BID	Placebo	gBRCAm PFS 11.2 vs. 4.3 m HR 0.18 (95% CI, 0.10–0.31)
SOLO-2 NCT01874353	III	Recurrent OC/FP/PPC ≥2 platinum-based chemotherapy With an objective response to the platinum regimen BRCAm	Olaparib 300 mg BID	Placebo	PFS 19.1 vs. 5.5 m HR 0.30 (95% CI, 0.22–0.41)
NOVA NCT01847274	III	Recurrent HGSOC/FP/PPC ≥2 platinum-based chemotherapy Platinum sensitive (>6 months)	Niraparib 300 mg daily	Placebo	gBRCAm PFS: 21.0 vs. 5.5 m HR 0.27 (95% CI, 0.17–0.41) Non-gBRCA PFS 9.3 vs. 3.9 m HR 0.45 (95% CI, 0.34–0.61)
ARIEL-3 NCT01968213	III	Recurrent HGSOC/endometrioid (or FP/PPC) ≥2 platinum-based chemotherapy Platinum sensitive (>6 months) ≤1 non-platinum chemotherapy CR/PR platinum-based chemotherapy	Rucaparib 600 mg BID	Placebo	ITT PFS 10.8 m vs. 5.4 m HR 0.37 (95% CI, 0.30–0.45) BRCAm PFS 16.6 m vs. 5.4 m HR 0.23 (95% CI, 0.16–0.34)

**Table 3 cancers-14-01235-t003:** Clinical trials of PARP inhibitors as maintenance in the front-line setting.

Study	Phase	Population	Study Arm	Control Arm	Results
SOLO-1 NCT01844986	III	HGSOC/endometrioid (or FP/PPC) FIGO III–IV BRCAm CR/PR platinum-based chemotherapy	Olaparib 300 mg BID	Placebo	PFS NR vs. 13.8 m HR 0.30 (95% CI 0.23–0.41)
PRIMA NCT02655016	III	HGSOC/endometrioid (or FP/PPC) FIGO III–IV Regardless of BRCA status CR/PR platinum-based chemotherapy	Niraparib 300 mg daily	Placebo	ITT PFS 13.8 vs. 8.2 m HR 0.62 (95% CI 0.50–0.76) HRD PFS 21.9 vs. 10.4 m HR 0.43 (95% CI 0.31–0.59) BRCAmut 0.40 (95% CI, 0.27–0.62)
PAOLA-1 NCT02477644	III	HGSOC/endometrioid/other epithelial non-mucinous (or FP/PPC) FIGO IIIB, IIIC or IV gBRCAm/BRCAwt if HGS CR/PR platinum-based chemotherapy	Olaparib 300 mg BID + bevacizumab 15 mg/kg/3 wks	Placebo + Bevacizumab 15 mg/kg/3 wks	ITT PFS 22.1 vs. 16.6 m HR 0.59 (95% CI 0.49–0.72) BRCAmut HR 0.31 (95% CI 0.20–0.47)
VELIA NCT02470585	III	HGSOC OC (or FP/PPC) FIGO III–IV	Paclitaxel-carboplatin-veliparib (150 mg BID-2 weeks 400 mg BID) → veliparib	Paclitaxel-carboplatin-placebo → placebo	ITT PFS: 23.5 vs. 17.3 m HR 0.68 (95% CI 0.56–0.83) BRCAmut PFS: 34.7 vs. 22 m HR 0.44 (95% CI 0.28–0.68)

**Table 4 cancers-14-01235-t004:** Clinical trials of PARP inhibitors as single agent therapy.

Study	Phase	Population	Study Arm	Control Arm	Results
SOLO-3 NCT02282020	III	Recurrent HGSOC/endometrioid (or FP/PPC) ≥2 platinum-based chemotherapy Platinum sensitive (>6 months) BRCAm	Olaparib 300 mg BID	Chemotherapy	PFS 13.4 vs. 9.2 m HR 0.62 (95% CI 0.43–0.91)
Study-10 NCT01482715	I/II	Recurrent HG OC/FP/PPC ≥3 platinum-based chemotherapy BRCAm	Rucaparib 600 mg BID	No comparator arm	ORR 59.5% mDOR 7.8 m (95% CI 5.6–10.5)
ARIEL-2 NCT01891344	II	Recurrent HGSOC (G2 or G3)/endometrioid (or FP/PPC) Prior platinum-based chemotherapy Platinum sensitive (>6 months) (R: 8 weeks from the last cycle)	Rucaparib 600 mg BID	No comparator arm	PFS BRCAm: 12.8 m LOH low: 5.2 m LOH high: 5.7 m HR 0.27 (95% CI 0.16–0.44)
ARIEL-4 NCT02855944	III	Recurrent HG OC/FP/PPC ≥2 chemotherapy regimens g/s BRCAm	Rucaparib 600 mg BID	Chemotherapy	PFS 7.4 m vs. 5.7 m HR 0.64 (95% CI 0.49–0.84)
QUADRA NCT02354586	II	Recurrent HGSOC (or FP/PPC) ≥3–4 previous chemotherapy regimens Platinum sensitive (>6 months) HRD/gBRCA testing	Niraparib 300 mg daily	No comparator arm	PFS 5.5 m (95% CI 3.5–8.2)

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
