# Peer review of "The BRCA Gene in Epithelial Ovarian Cancer"

_cancers, 2022, doi:10.3390/cancers14051235_

Round 1

Reviewer 1 Report

Well written review

Author Response

Dear reviewer, we would like to thank you for your time and your comments on our review article.

Reviewer 2 Report

The aim of this manuscript is to evaluate the impact of BRCA gene in epithelial ovarian cancer, not only in the pathogenesis of epithelial ovarian cancer, but also as an essential element in the homologous recombination system, for DNA double-strand break repair.

Even if the manuscript provides an organic overview, with a densely organized structure and based on well-synthetized evidence, there are aspects to be mentioned, to make the article fully readable. For these reasons, the manuscript requires minor changes.

Please find below an enumerated list of comments on my review of the manuscript:

LINE 22 – 24: Base excision repair (BER) corrects forms of single base damage, not perceived as a significant distorsion to the DNA helix. Nucleotide excision repair (NER),instead, repairs  multiple and bulky base damage, while mismatch repair (MMR)  is an evolutionarily conserved, post replicative repair pathway, which contributes to replication fidelity. In this context, the manuscript may benefit from providing a brief and organic explanation of the most significant mechanism of DNA Damage Repair (see, for reference: Chatterjee N, Walker GC. Mechanisms of DNA damage, repair, and mutagenesis. Environ Mol Mutagen. 2017 Jun;58(5):235-263. doi: 10.1002/em.22087. Epub 2017 May 9. PMID: 28485537; PMCID: PMC5474181).

LINE 101 – 103: Ovarian cancer disease inlcudes a heterogenous group of neoplasia: among them, about 90% are epithelial (subtypes: mucinous, serous, endometrioid and clear cells). Most of the women are diagnosed with high – grade serous ovarian cancer subtype, more aggressive and with a poorer prognosis, as highlighted by several morphological and ultrastructural studies (see, for reference: Giusti, I., Bianchi, S., Nottola, S. A., Macchiarelli, G., Dolo, V. (2019). CLINICAL ELECTRON MICROSCOPY IN THE STUDY OF HUMAN OVARIAN TISSUES. EuroMediterranean Biomedical Journal14).

In conclusion, this manuscript is densely presented and well organized, based on well-synthetized evidences. The authors were lucid in their style of writing, making it easy to read and understand the message, portrayed in the manuscript. Besides, the methodology design was rigorous and appropriately implemented within the study. However, many of the topics are very concisely covered. This manuscript provided a comprehensive review of current knowledge in this field. Moreover, this research have futuristic importance and could be potential for future research. However, I have minor comments only for the introductive section, for improvement before acceptance for publication. The article is accurate and provides relevant information on the topic and I suggest minor changes to be made in order to maximize its scientific impact. I would accept this manuscript, if the comments are addressed properly.

Author Response

Reviewer 2: Dear reviewer, we greatly appreciate your comments on our paper and agree that these changes will enhance our review article.  We have taken your comments into consideration and have made the following changes to our manuscript: 

-We made a brief explanation of the most significant mechanism of DNA damage repair adding the reference you gently provided

The human cells (in general all eukaryotic cells) have developed different pathways to fix the different types of DNA damage, either affecting one (SSBs, single strand breaks) or both (DSBs, double strand breaks) DNA strands. When only one DNA strand is broken, and thus the other is available as a template, it can be repaired by base excision repair (BER), nucleotide excision repair (NER) or mismatch repair (MMR) (2). BER corrects forms of single base damage, not perceived as a significant distorsion to the DNA helix.  NER, instead, repairs  multiple and bulky base damage, while MMR is an evolutionarily conserved, post replicative repair pathway, which contributes to replication fidelity (3). If both strands are involved, the pathways available for repair are non-homologous end joining (NHEJ) and homologous recombination (HR), which includes gene conversion (GC) and single strand annealing (SSA) (2). NHEJ is critical for the repair of pathologic DSBs, chromosomal translocations and also for the repair of physiologic DSBs created during V(D)J recombination and class switch recombination . HR utilizes DNA strand invasion and template-directed DNA repair synthesis to effect a high-fidelity repair (3).

-We tried to make clear the information you remarked and added the reference you gently provided.

Although, several founder mutations of the BRCA and BRCA2 genes have been reported, some of these mutation carriers have a different risk of developing cancer. An example of this has been the identification of an ovarian cancer cluster region (OCCR) in or near exon 11 in BRCA1 and BRCA2. The Consortium of Investigators of Modifiers of BRCA1/2 (CIMBA) revealed that the incidence of OC is high in patients with germline BRCA mutation in the OCCR in about 30,000 BRCA mutant carriers in 33 countries around the world. Pathogenic variants within the OCCR have been associated with a higher ratio of ovarian to breast cancer (12).

Reviewer 3 Report

The paper by Lorenzo et al, is a very simple overview of the role of BRCA in ovarian cancer. Overall the review presents a nice summary but novel new information is not presented and the aim of the review ie gap in knowledge, t is not clearly stated. This review would also benefit from extensive English proofreading for sentence structure and wording.

Minor corrections

  • Gene names should always be in italics
  • Yellow text (STOP) in Figure 1 is hard to read. Please change to another colour
  • Line 88: Full stop required after reference 9.
  • Line 94: Please change “recommend” to “recommended for
  • Line 136: Change “associated to a better prognosis” to “associated with a better prognosis”
  • Line 137: Change “studied from years” to “studied for years”
  • Line 153: Correct BRCA 1 to BRCA1
  • Line 163 and 168: Correct “metanalysis” to “meta-analysis”. Please correct throughout text.
  • Line 164. Change correlate to correlated.
  • Line 165. Remove “the” at the end of the sentence.
  • Line 177: Remove space after “of”
  • Drug names: Sometime drugs are written with a capital letter ie Olaparib Line 241, others are written as olaparib (Line 238). Please keep consistent throughout text

BRCA gene in the Clinic section

  • A better understanding of the BRCA mutations observed in ovarian cancer should be included. Are BRCA mutations more common in some ovarian subtypes compared to others? What are the main mutations found in ovarian cancer, ie are there hot spot mutations? How do these mutations effect the efficacy of PARP inhibitors? Do different BRCA mutations confer better overall survival?

Author Response

Dear reviewer, we greatly appreciate your comments on our paper and agree that these changes will enhance our review article.  We have taken your comments into consideration and have made the following changes to our manuscript: 

-The gene names have been italicized, as you remarked that is the correct nomenclature.

-We have changed the “STOP” in Figure 1 to blue color, as you remarked yellow color was hard to read.

-The typos that you gently highlighted have been corrected and we tried to review the sentence structure.

-We have corrected the drug names starting all of them with a capital letter to keep consistency throughout the text.

-About the comments for the BRCA gene in the clinic section:

            -We tried to address the most common ovarian cancers subtypes in section 2. BRCA gene and heredity ovarian cancer syndrome

            -About the OS difference in BRCA1 and BRCA2 mutation carriers we tried to introduce the topic with the Zhong meta-analysis. We have included additional information trying to clarify the different clinical effects of BRCA1 and 2 mutations, lines 249-253.

            -We added an additional paragraph about the efficacy difference of PARPi between BRCA1 and BRCA2 mutation carriers, and a possible explanation for this lack of difference at least with the information that we have so far, lines 291-306.

            -We included a paragraph about the ovarian cancer cluster region identified in BRCA1 and 2 in order to address the comments about the main mutations found in ovarian cancer as well as the one about a better understanding of the BRCA mutations observed in ovarian cancer, lines 108-121.

Reviewer 4 Report

The authors well describes the importance of BRCA test at the primary diagnosis of patients with Epithelial ovarian cancer (EOC).

The review describe in details the implication of the gene in the clinic and in therapeutic 

Author Response

(The authors gave the same response as above.)

Round 2

Reviewer 3 Report

Well done on the resubmission. Unfortunately the authors have not addressed two questions that were raised during the review process.

  1. Are BRCA mutations more common in some ovarian subtypes compared to others?
  2.  What are the main mutations found in ovarian cancer, ie are there hot spot mutations?

Author Response

Dear reviewer, we greatly appreciate your comments on our paper and we are happy that you are satisfied with some of the changes that we made after your suggestions. 

We have taken your comments into consideration and have made the following changes to our manuscript: 

-We have tried to better describe the most common subtypes in which you can find a BRCA1 and/or BRCA2 mutation and also how frequent or infrequent are the mutations in the other subtypes (lines 143-146). 

-We have described the location of the ovarian cancer cluster regions identified in BRCA1(OCCR1) and BRCA2 (OCCR1 and OCCR2) to better address your comments, and we apologize for not being that precise before (lines 126-129). 
